# ReCODE: A Personalized, Targeted, Multi-Factorial Therapeutic Program for Reversal of Cognitive Decline

**DOI:** 10.3390/biomedicines9101348

**Published:** 2021-09-29

**Authors:** Rammohan V Rao, Sharanya Kumar, Julie Gregory, Christine Coward, Sho Okada, William Lipa, Lance Kelly, Dale E Bredesen

**Affiliations:** 1Apollo Health, Burlingame, CA 94011, USA; julie@ahnphealth.com (J.G.); chris@ahnphealth.com (C.C.); sho@ahnphealth.com (S.O.); bill@ahnphealth.com (W.L.); lance@ahnphealth.com (L.K.); 2Voluntary Data Scientist, Chennai 600083, India; sharanyakumar6@gmail.com; 3Department of Molecular and Medical Pharmacology, University of California, Los Angeles, CA 90024, USA

**Keywords:** Alzheimer’s disease, cognitive decline, therapeutics, diet, lifestyle, supplements, blood analysis, AD risk factors

## Abstract

Background: Alzheimer’s disease (AD) is the major cause of age-associated cognitive decline, and in the absence of effective therapeutics is progressive and ultimately fatal, creating a dire need for successful prevention and treatment strategies. We recently reported results of a successful proof-of-concept trial, using a personalized, precision medicine protocol, but whether such an approach is readily scalable is unknown. Objective: In the case of AD, there is not a single therapeutic that exerts anything beyond a marginal, unsustained, symptomatic effect. This suggests that the monotherapeutic approach of drug development for AD may not be an optimal one, at least when used alone. Using a novel, comprehensive, and personalized therapeutic system called ReCODE (reversal of cognitive decline), which proved successful in a small, proof-of-concept trial, we sought to determine whether the program could be scaled to improve cognitive and metabolic function in individuals diagnosed with subjective cognitive impairment, mild cognitive impairment, and early-stage AD. Methods: 255 individuals submitted blood samples, took the Montreal Cognitive Assessment (MoCA) test, and answered intake questions. Individuals who enrolled in the ReCODE program had consultations with clinical practitioners, and explanations of the program were provided. Participants had follow-up visits that included education regarding diet, lifestyle choices, medications, supplements, repeat blood sample analysis, and MoCA testing between 2 and 12 months after participating in the ReCODE program. Pre- and post-treatment measures were compared using the non-parametric Wilcoxon signed rank test. Results and Conclusions: By comparing baseline to follow-up testing, we observed that MoCA scores either significantly improved or stabilized in the entire participant pool—results that were not as successful as those in the proof-of-concept trial, but more successful than anti-amyloid therapies—and other risk factors including blood glucose, high-sensitivity C-reactive protein, HOMA-IR, and vitamin D significantly improved in the participant pool. Our findings provide evidence that a multi-factorial, comprehensive, and personalized therapeutic program designed to mitigate AD risk factors can improve risk factor scores and stabilize or reverse the decline in cognitive function. Since superior results were obtained in the proof-of-concept trial, which was conducted by a small group of highly trained and experienced physicians, it is possible that results from the use of this personalized approach would be enhanced by further training and experience of the practicing physicians. Nonetheless, the current results provide further support indicating the potential of such an approach for the prevention and reversal of cognitive decline.

## 1. Introduction

Alzheimer’s disease (AD) is the major cause of age-related cognitive decline, with approximately 6.2 million Americans age 65 and older estimated to suffer from this disease [1]. With 13 million Americans and 160 million people globally projected to have AD by 2050, the impact it will have on healthcare systems worldwide requires serious consideration [1,2,3]. The cause(s) of AD remain unclear, and there is no truly effective treatment currently recognized. This makes the need to develop effective prevention and treatment increasingly pressing. Recent studies support the notion that metabolic abnormalities are present in patients with cognitive decline, often years prior to a diagnosis of AD [4,5]. Studies also suggest the effect of metabolic abnormalities, such as insulin resistance, chronic inflammation, hypovitaminosis D, hormonal deficiencies, and hyperhomocysteinemia, among others, in the AD process [5,6,7,8,9].

Thus, a therapeutic strategy to identify and attenuate all the risk factors specific to each affected individual may have a significant impact on disease progression, as has been shown recently [10,11,12,13,14]. Here we present pilot data that a comprehensive and personalized therapeutic program designed to mitigate AD risk factors can improve several risk factor scores and stabilize cognitive function, warranting prospective, longitudinal cohort studies, and controlled clinical trials.

## 2. Methods

### 2.1. Study Design and Participant Enrollment

ReCODE is a comprehensive and personalized multi-therapeutic program for reversing symptoms of cognitive decline and optimizing brain health, using a targeted algorithm based on biochemical and genetic risk factors for cognitive decline. It is intended for individuals experiencing symptoms of subjective cognitive impairment (SCI), mild cognitive impairment (MCI), and those with early stage AD, although some with later stages of AD have shown improvement, as well [13]. The ReCODE program includes information on the metabolic factors that drive the symptoms of cognitive decline and provides detailed, personalized recommendations to address these factors, such as nutrition, exercise (physical and mental), sleep, stress management, detoxification, supplements, and hormones. The ReCODE program evolved from other similar programs that used precision medicine approaches to identify and target the drivers of Alzheimer’s or pre-Alzheimer’s [13,14,15]. Following the completion of labs, medical questionnaires, and cognitive testing, a software-based algorithm generates a personalized report that addresses the identified putative contributors to cognitive decline, such as specific pathogens, toxins, or hormonal alterations.

Patients who received a diagnosis of SCI, MCI, or early stage AD from their practitioners and who chose to enroll in the ReCODE program were included. Patients who had any major medical illnesses, such as cardiovascular disease or cancer, or who received a psychiatric diagnosis that impacted cognition, as well as pregnant women, were excluded. Patients work closely with a ReCODE-trained physician, health coach, nutritionist, and other practitioners as needed. Their protocol is integrated with all members of the clinical team, allowing for easy access, communication, and support. Patient data include AD symptoms, demographics, past and current medical history, physical and mental health history, diet and lifestyle patterns, tobacco, alcohol, recreational drug usage, current medications and supplement usage, family history, social history, and current living environment.

Patients undergo genetic and blood testing for a range of parameters associated with the onset of AD. A subset of those components is listed in Table 1.

Data are then processed with the ReCODE software, which identifies the potential underpinning contributors to cognitive decline, or risk for decline, for each person. Data are then screened by a ReCODE-trained physician and a health care team, supporting them in their recommendation of a personalized, therapeutic intervention. Participants are encouraged to work with a health coach for support and adherence to the program to halt progression and reverse symptoms of cognitive decline while optimizing overall health.

### 2.2. The ReCODE Program

The program is driven by a data set that is much larger than the standard-of-care dementia evaluation data set, in order to identify the upstream contributors to cognitive decline, such as various pathogens, gut dysbiosis, sleep disorders, various toxins, hormonal deficiencies, and other potential contributors. A report is then generated that results from the culmination of data (lab work, medical questionnaires, and cognitive testing results) and run through a computer algorithm that provides in-depth analysis about potential contributors—inflammation, insulin resistance, nutrient and hormonal deficiencies, specific pathogens, toxicants, and biotoxins, as well as genetics. The participant is then provided with suitable information for addressing and identifying each contributor with his/her physician. In all participants, combinations of multiple contributors driving their disease processes were identified; in none was a single contributor identified.

In addition to addressing each contributor, the participants were advised to implement seven foundational diet and lifestyle strategies that provided a basis for healing. The strategies involve the use of a plant-rich (but not plant-exclusive) ketogenic diet that prioritizes low mercury, wild-caught seafood, and pastured eggs, a long daily fast; an exercise program comprised of both aerobic and anaerobic exercise combined with the suggestion to avoid prolonged periods of sitting; 7–8 h of quality, restorative sleep through the adoption of a sleep hygiene program (and treatment of nocturnal hypoxemia if present); a stress management program emphasizing regular deep breathing breaks with meditation; regular brain training, other learning opportunities, as well as the maintenance of social connectivity; an avoidance of toxins along with instructions to upregulate detoxification; and, finally, personalized supplement recommendations based upon the participant’s lab values.

### 2.3. Statistical Analysis

Participants were pooled, and for statistical analysis pre- and post-treatment groups were compared. To visualize the datasets of various parameters from human samples, we used the boxplot representation method and generated box and whisker plots for each group of datasets. While the total sample size was 255, the participant pool differs in each test parameter since each test parameter was first analyzed by the interquartile range (IQR) method to identify and discard any confounding outliers. The median and IQR were computed for both pre-treatment and post-treatment groups. In all the boxplots, the ends of the box are the upper and lower quartiles, so the box spans the interquartile range, the horizontal line within the box represents the median value, the whiskers are the two lines outside the box that extend to the highest and lowest observations, and the small square represents mean value. All the boxplots with the median as the center value provide a brief picture of the other important distribution values. Pre- and post-treatment measures were compared using the nonparametric Wilcoxon signed-rank test (primary analysis approach). A *p*-value < 0.05 was considered significant.

## 3. Results

Data from 255 people with at least two or more follow-up blood panels and the Montreal Cognitive Assessment (MoCA) tests to allow for statistical analysis of efficacy are presented. Statistics on the participant pool are shown in Table 2.

In general, participants did not have other co-morbid conditions, and about 70% of them possessed at least one ApoE4 allele, a major genetic risk factor for AD development [9,16,17]. Biochemical and cognitive tests were performed at enrollment and at intervals of three months after beginning treatment. The target ranges for the parameters reported in this study are shown in Table 3.

Results are expressed as two groups, including the pre-treatment group and 2–12 months follow-up. Within the treatment group, if any given participant had more than one value, we utilized the mean value for the analysis to determine statistical relevance.

The ReCODE program was developed to offset the range of several biochemical risk factors linked to AD, making it possible to test whether these parameters can be altered. In addition to being a risk factor of cardiovascular disease, elevated levels of high-sensitivity C-reactive protein (hs-CRP) are suggested to serve as a biomarker for cognitive impairment risk associated with systemic inflammation [18,19,20,21]. As shown in Figure 1, a significant decline in hs-CRP was observed in the treatment group compared to the high levels at baseline (Figure 1a).

Vitamin D (Vit D) is one of the most important nutrients supporting brain function, and reduced Vit D levels are associated with cognitive decline [21,22,23]. A significant improvement in Vit D levels was observed in the treatment group. Vit D, which was below 40 ng/mL in 44% of subjects prior to treatment, showed a significant increase in the group as a whole following treatment (Figure 1b).

Glycotoxicity triggers both inflammation and insulin resistance, and thus contributes to AD risk [24,25,26]. One of the methods to assess insulin resistance is to assess both fasting glucose and insulin levels and calculate the Homeostatic Model Assessment for Insulin Resistance (HOMA-IR) [13,27]. Prior to undertaking the ReCODE program, the HOMA-IR value was elevated in the pretreatment group (Figure 2a). Following the treatment protocol, a significant decline in HOMA-IR was observed in participants (Figure 2a). Participants also had high glucose levels prior to the treatment, which improved significantly after adopting the protocol (Figure 2b).

MCI and AD are characterized by a steady decline in cognitive performance that can be detected by using cognitive assessment tools, such as the Montreal Cognitive Assessment (MoCA), which has been well accepted as a relatively sensitive and efficient screening tool [28,29]. MoCA test scores from participants prior to and after the inception of the program are shown in Figure 3. In Figure 3a, both baseline and post-treatment MoCA scores ranged from 1 to 30 with a mean of 21. Although 45% of the total subjects (*n* = 251) showed improvement in the post-treatment scores, statistical significance was not achieved. However, since ~20% of the total subjects had baseline MoCA scores ≤ 9, we reanalyzed the data by including only those subjects that had MoCA scores ≥ 10. In Figure 3b, both baseline and post-treatment MoCA ranged from 11 to 30 with a mean of 23. While the baseline median score was 23, the post-treatment median score was 24, and improvement in the post-treatment scores was statistically significant (*p* < 0.05).

A MoCA score of 19–26 is indicative of a mild but noticeable decline in cognition. A re-analysis of the data by including only those subjects with MoCA scores ≥ 19 (*n* = 151) demonstrated a statistically significant improvement (*p* < 0.005), as shown in Figure 3c. The range of MoCA changes in this group was from −9 (28 to 19) to +8 (19 to 27, 20 to 28, 21 to 29 and 22 to 30). Out of the 151 subjects, 76 (50%) improved their scores, 41 (27%) showed a decline in scoring, and 34 (23%) were unchanged. For comparison, in the proof-of-concept trial using this same approach, 76% of the patients showed an improvement in MoCA scores at the end of the 9-month trial [14]. The current results, together with the metabolic evaluations, demonstrate improvement that is highly significant statistically.

These overall findings indicate that participants in the ReCODE program experienced improved metabolic parameters and cognition. Although not shown here, a subset of other risk factors that improved by similar strategies included several hormones, vitamin B12, homocysteine, and serum zinc, supporting the notion that the ReCODE strategy to effectively mitigate metabolic risk factors may result in arresting cognitive decline or improving cognitive performance [13,14].

## 4. Discussion & Conclusions

In the past decade alone, hundreds of clinical trials have been conducted for AD at an aggregate cost of billions of dollars, without significant success, when success is defined as cognitive improvement, as opposed to a slowing of decline [1,2,3]. Extensive resources have been directed toward drug trials, the failure of which has led some to question whether the monotherapeutic approach for SCI, MCI, or AD is an optimal one [4]. While effective drugs may be developed, other approaches, such as treating risk factors, may delay the disease onset or slow progression. Recent studies suggest that dietary and other lifestyle changes are the most effective way currently to prevent, slow, or reverse AD disease progression [4,10,11]. These studies demonstrate that a personalized, comprehensive, systems-based precision approach offers potential for cognitive improvement, and raises the question of whether combining pharmaceuticals with personalized protocols may improve overall outcomes.

The ReCODE program is a multifactorial approach that identifies and targets the potential contributors to cognitive decline in each patient. It involves seven core strategies to increase resiliency and optimize brain function to prevent or delay AD symptoms in addition to the personalized targeting of potential contributors, such as specific pathogens, nocturnal hypoxemia, gastrointestinal hyperpermeability, altered gut microbiome, altered oral microbiome, altered immunity, and other factors. While each of the strategies—nutrition, physical exercise, sleep, stress, brain stimulation, detoxification, and targeted supplementation—may support cognitive health to some extent, when practiced together, they create synergy. Although there are other studies evaluating the benefits of exercise, vitamin deficiencies, sleep, mental fitness, and glucose metabolism on the reversal of cognitive decline, these single interventions may not lead to sustainable improvement since each alone may not target the underlying drivers of the degenerative process. Our prior research showed that if all the core strategies are implemented and followed simultaneously, and the program continued, the improvements from these strategies are typically sustained, thus offering a major advantage over monotherapeutics [8,13,15,21,30]. Additionally, if the symptoms and potential causes for AD are identified at the early pre-symptomatic stages, the implementation of ReCODE allows for correcting these contributors and preserving cognitive health.

While our primary goal was to recruit participants that were in the early stages of disease (SCI, MCI, and early-moderate AD), there were several participants who were in more advanced stages of AD, and these more advanced patients did not demonstrate the cognitive improvement observed in those in earlier stages. Metabolic factors, including fasting serum glucose, HOMA-IR, vitamin D, and hs-CRP, improved significantly in the group as a whole; however, it is not yet clear whether this enhancement is required for the cognitive improvement that was observed in those with MoCA ≥ 10.

The MoCA is a widely used screening assessment for detecting cognitive impairment. Scores on the MoCA range from zero to 30, with a score of 26 and above generally considered normal, a score range of 19–25 (average = 22) compatible with MCI, and a score range of 11–21 (average = 16) compatible with moderate to mild AD [31,32]. In the analysis of the entire group, the improvement failed to reach statistical significance. However, those with MoCA scores ≥ 10 showed improvement that was statistically significant. Furthermore, approximately 50% of the participants that had MoCA scores ≤ 9 and who would be categorized as being in the severe stage of AD showed significant improvement in their biochemical tests despite showing no improvement in their MoCA scores. This suggests that the putative causal biochemical factors, such as insulin resistance, vitamin D, and others, may be more amenable to improvement compared to MoCA, which may require a longer adherence to the treatment program. The gradual improvement in MoCA scores, especially for those subjects in the moderate-severe stage of AD, may also require strict compliance to the program in order to improve and sustain the therapeutic benefits.

Although the current study documented significant cognitive benefits in those with MoCA scores ≥ 10, neither the magnitude of the improvements nor the fraction of patients who improved matched the results obtained in the recent proof-of-concept trial that employed the same approach [14]. This held true even when the evaluation in the current study was restricted to those with MoCA scores ≥ 19 (which matched the cognitive scores in the trial). This indicates that the protocol did not scale completely, suggesting that there may be a physician-experience component in the efficacy of this approach.

While this study focused on four metabolic factors and one cognitive test, the ReCODE program involves more than 50 factors contributing to cognitive decline. It will be interesting to determine how the other factors influence the disease progression as more data become available. In summary, our findings from this pilot study reinforce the potential benefits of a multitherapeutic, personalized approach targeting metabolic risk factors and support our conclusion that the ReCODE program may be beneficial in delaying or arresting poor cognitive performance in most cases. The results also pave the way for larger controlled studies to provide further validation.

## Figures and Tables

**Figure 1 biomedicines-09-01348-f001:**
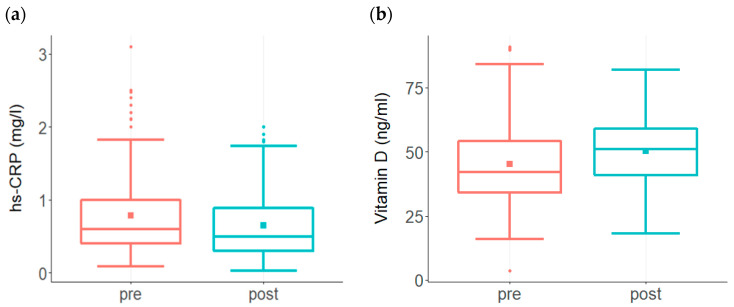
Changes in risk factor levels among participants enrolled in the ReCODE program. (**a**) A significant decline in hs-CRP (*p* < 0.001) was observed in the treatment group (*n* = 177) compared to the high levels at baseline. While baseline hs-CRP ranged from 0.09 to 3.1 with a mean of 0.8, post-treatment levels ranged from 0.03 to 2 with a mean of 0.64. The baseline median score was 0.6, post-treatment median score was 0.5 (**b**) Baseline Vit D levels among participants (*n* = 207) ranged from 3.6 to 91 with a mean of 45 and median score of 42, post-treatment levels ranged from 18 to 82 with a mean of 50 and median score of 52. This increase in the post-treatment group was statistically significant (*p* < 0.001).

**Figure 2 biomedicines-09-01348-f002:**
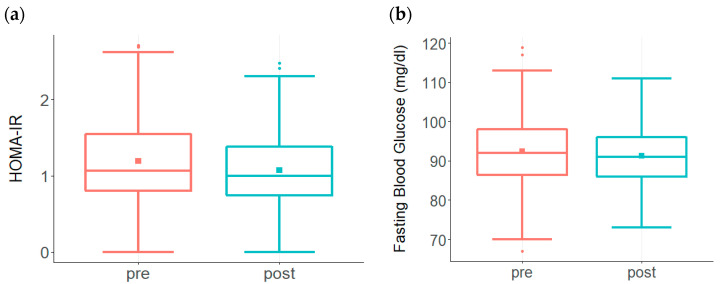
Changes in HOMA-IR and Fasting Glucose levels among participants enrolled in the ReCODE program. (**a**) A significant decline in HOMA-IR (*p* < 0.002) was observed in the treatment group (*n* = 176) compared to the high levels at baseline. While baseline HOMA-IR ranged from 0 to 2.7 with a mean of 1.2, post-treatment levels ranged from 0 to 2.5 with a mean of 1.07. The baseline median score was 1.1, post-treatment median score was 1.0. (**b**) A significant reduction in fasting glucose (*p* < 0.01) is observed among participants (*n* = 208) following the treatment protocol. While baseline fasting glucose levels ranged from 67 to 119 with a mean of 93, post-treatment scores ranged from 73 to 111 with a mean of 91. The baseline median score was 92, post-treatment median score was 91.

**Figure 3 biomedicines-09-01348-f003:**
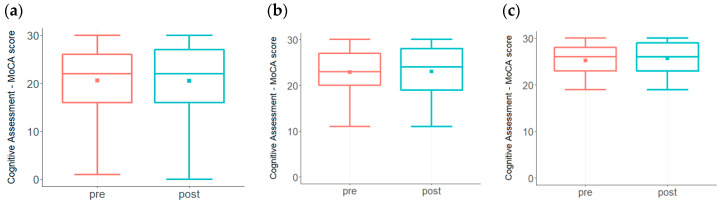
MoCA test scores from participants prior to and after the inception of the program. While there is a trend toward increased scores (**a**) among the entire participant pool (*n* = 251), statistical significance was not achieved (*p* = 0.484). (**b**). MoCA test scores were evaluated among participants with baseline score ≥ 10 (*n* = 212). Improvement in the post-treatment scores was statistically significant (*p* < 0.05). (**c**). MoCA test scores of participants (*n* = 151) with a baseline score of ≥ 19 showed statistically significant improvement in the post-treatment scores (*p* < 0.005).

**Table 1 biomedicines-09-01348-t001:** A subset of ReCODE evaluation panel components.

Categories	Parameter
General Health	BMI
Assessments	MoCA score
Inflammation	A/G Ratio, IL-6, TNF-Alpha, hs-CRP
Methylation	Folate, Homocysteine, Vitamin B12, Vitamin B6
Hormones	Cortisol, DHEA-Sulfate, Estradiol, T3, T4, TSH, Testosterone, Pregnenolone, Progesterone, Vitamin D
Metals	Potassium, Magnesium, Copper, Zinc
Others	Lipid panel, Chemistry panel, Toxins (Organics and Heavy Metals), Cytoprotection, Antioxidants, Mycotoxins

**Table 2 biomedicines-09-01348-t002:** Study Participants-Demographics.

Total Subjects	*n* = 255
Males	111
Females	144
Average age	73
ApoE4-absent	89
ApoE4-1 copy	104
ApoE4-2 copies	40

**Table 3 biomedicines-09-01348-t003:** Target ranges of the parameters reported in this study.

Parameter	Target Range
MoCA	26–30
hs-CRP	<0.9 mg/L
Vitamin D	40–80 ng/mL
HOMA-IR	≤1.2
Fasting Glucose	70–90 mg/dL

## Data Availability

Not Applicable.

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
