# Peer review of "ReCODE: A Personalized, Targeted, Multi-Factorial Therapeutic Program for Reversal of Cognitive Decline"

_biomedicines, 2021, doi:10.3390/biomedicines9101348_

Round 1
Reviewer 1 Report
The manuscript by Rao et al.,. presents a novel, comprehensive, and personalized therapeutic system called Re-CODE (reversal of cognitive decline), which proved successful in a small, proof-of-concept trial, and report that the program could be scaled to improve cognitive and metabolic function in individuals diagnosed with subjective cognitive impairment, mild cognitive impairment, and early-stage AD. The Authors provide evidence that a multi-factorial, comprehensive, and personalized therapeutic program designed to mitigate AD risk factors can improve risk factor scores and stabilize or reverse the decline in cognitive function.
The review is of interest and well-designed.
Minor comment: Table 1: "Title 1" "Title 2" have to be corrected
Author Response
Reviewer 1.
- The manuscript by Rao et al., presents a novel, comprehensive, and personalized therapeutic system called Re-CODE (reversal of cognitive decline), which proved successful in a small, proof-of-concept trial, and report that the program could be scaled to improve cognitive and metabolic function in individuals diagnosed with subjective cognitive impairment, mild cognitive impairment, and early-stage AD. The Authors provide evidence that a multi-factorial, comprehensive, and personalized therapeutic program designed to mitigate AD risk factors can improve risk factor scores and stabilize or reverse the decline in cognitive function.
The review is of interest and well-designed. Minor comment: Table 1: "Title 1" "Title 2" have to be corrected
Authors Reply: Thanks for your positive and highly encouraging comments. We have corrected Table 1: "Title 1" "Title 2"

Reviewer 2 Report
The manuscript entitled „ReCODE: A personalized, targeted, multi-factorial therapeutic 2 program for reversal of cognitive decline” by Rao et al., is well written and of interest to physicians and especially large public since it turns the attention from one single treatment to a more complex one in neurodegeneration. The authors highlighted the effect of metabolic abnormalities such as insulin resistance, chronic inflammation, hypovitaminosis D, hormonal deficiencies, and hyperhomocysteinemia, among others in the AD process, which is a new appoach in AD. Therefore, they reffer to „a comprehensive and personalized 59 therapeutic program designed to mitigate AD risk factors can improve several risk factor 60 scores and stabilize cognitive function, warranting prospective, longitudinal cohort stud- 61 ies and controlled clinical trials.”
I would suggest to the authors to take into consideration for their next research the effect of food, and especially the blood glucose on the progression of AD. The life style may be of great significance for degenerative diseases, such as nutrition (too much food daily), lack of physical exercises, anxiety, etc. How to measure all these to have a link with AD?
In principle, the manuscript can be published as it is.
Author Response
Reviewer 2.
The manuscript entitled ReCODE: A personalized, targeted, multi-factorial therapeutic 2 program for reversal of cognitive decline” by Rao et al., is well written and of interest to physicians and especially large public since it turns the attention from one single treatment to a more complex one in neurodegeneration. The authors highlighted the effect of metabolic abnormalities such as insulin resistance, chronic inflammation, hypovitaminosis D, hormonal deficiencies, and hyperhomocysteinemia among others in the AD process, which is a new approach in AD.
Therefore, they refer to a comprehensive and personalized therapeutic program designed to mitigate AD risk factors can improve several risk factor scores and stabilize cognitive function, warranting prospective, longitudinal cohort studies and controlled clinical trials.
I would suggest to the authors to take into consideration for their next research the effect of food, and especially the blood glucose on the progression of AD. The life style may be of great significance for degenerative diseases, such as nutrition (too much food daily), lack of physical exercises, anxiety, etc. How to measure all these to have a link with AD?
In principle, the manuscript can be published as it is.
Authors Reply: Thanks for your positive and highly encouraging comments. We are in total agreement with you regarding your last comment about future studies. We have already embarked on the next phase of the study looking at the effect of diet, exercise and mental status on the progression/reversal of AD. Our outcome measures include several factors implicated in AD pathogenesis including but not limited to hemoglobin A1c (glycotoxicity), TSH levels (Thyroid function is critical for brain health and cognition, and low thyroid is another contributor to type 2-atrophic Alzheimer’s disease), Glutathione (contributes to inflammation, toxicity, and loss of support for synapses) and CNS Vital Signs (neurocognitive testing is a non-invasive clinical procedure to efficiently and objectively assess a broad-spectrum of brain function domain performances).
